# A Hormetic Spatiotemporal Photosystem II Response Mechanism of Salvia to Excess Zinc Exposure

**DOI:** 10.3390/ijms231911232

**Published:** 2022-09-23

**Authors:** Michael Moustakas, Anelia Dobrikova, Ilektra Sperdouli, Anetta Hanć, Ioannis-Dimosthenis S. Adamakis, Julietta Moustaka, Emilia Apostolova

**Affiliations:** 1Department of Botany, Aristotle University of Thessaloniki, 54124 Thessaloniki, Greece; 2Institute of Biophysics and Biomedical Engineering, Bulgarian Academy of Sciences, 1113 Sofia, Bulgaria; 3Institute of Plant Breeding and Genetic Resources, Hellenic Agricultural Organisation-Demeter (ELGO-Demeter), 57001 Thermi, Greece; 4Department of Trace Analysis, Faculty of Chemistry, Adam Mickiewicz University, 61614 Poznań, Poland; 5Section of Botany, Department of Biology, National and Kapodistrian University of Athens, 15784 Athens, Greece

**Keywords:** hormesis, leaf heterogeneity, nutrient imbalances, chlorophyll fluorescence imaging, reactive oxygen species (ROS), photoprotection, non-photochemical quenching (NPQ), stress effects, chlorophyll, phenolics

## Abstract

Exposure of *Salvia sclarea* plants to excess Zn for 8 days resulted in increased Ca, Fe, Mn, and Zn concentrations, but decreased Mg, in the aboveground tissues. The significant increase in the aboveground tissues of Mn, which is vital in the oxygen-evolving complex (OEC) of photosystem II (PSII), contributed to the higher efficiency of the OEC, and together with the increased Fe, which has a fundamental role as a component of the enzymes involved in the electron transport process, resulted in an increased electron transport rate (ETR). The decreased Mg content in the aboveground tissues contributed to decreased chlorophyll content that reduced excess absorption of sunlight and operated to improve PSII photochemistry (Φ*_PSII_*), decreasing excess energy at PSII and lowering the degree of photoinhibition, as judged from the increased maximum efficiency of PSII photochemistry (F*v*/F*m*). The molecular mechanism by which Zn-treated leaves displayed an improved PSII photochemistry was the increased fraction of open PSII reaction centers (q*p*) and, mainly, the increased efficiency of the reaction centers (F*v*′/F*m*′) that enhanced ETR. Elemental bioimaging of Zn and Ca by laser ablation–inductively coupled plasma–mass spectrometry (LA–ICP–MS) revealed their co-localization in the mid-leaf veins. The high Zn concentration was located in the mid-leaf-vein area, while mesophyll cells accumulated small amounts of Zn, thus resembling a spatiotemporal heterogenous response and suggesting an adaptive strategy. These findings contribute to our understanding of how exposure to excess Zn triggered a hormetic response of PSII photochemistry. Exposure of aromatic and medicinal plants to excess Zn in hydroponics can be regarded as an economical approach to ameliorate the deficiency of Fe and Zn, which are essential micronutrients for human health.

## 1. Introduction

Zinc (Zn), an essential micronutrient for plants, is found in plant cells in its oxidized form, which is a divalent cation Zn^2+^ that is required for normal plant growth and development [1,2,3]. It is involved in various essential cell functions, retaining, among others, the membranous structure of various cell organelles [4,5,6,7,8,9]. Zn is also a significant structural component of many proteins involved in numerous biological processes, including the metabolism of reactive oxygen species (ROS), photosynthesis, transcription, and translation [3,10,11]. Zinc cations and Zn complexes have antioxidant and antimicrobial activities [12].

Some plants, known as metallophytes, are Zn hyper-tolerant through Zn exclusion and detoxification mechanisms, while others vigorously accumulate extraordinarily high Zn concentrations [11,13]. Leaf Zn concentrations between 30 and 100 μg Zn^2+^ g^−1^ dry weight (DW), are considered optimal, while concentrations above 300 μg Zn^2+^ g^−1^ DW (0.03% dry mass) are considered toxic to plants [1,14]. However, plant species that are extremely tolerant to Zn can hyperaccumulate more than 10,000 μg Zn^2+^ g^−1^ DW without displaying toxicity symptoms [15]. Zn-toxicity symptoms include growth stunting, oxidative stress, and chlorosis [1].

Environmental stress conditions, including heavy metals, result in oxidative stress induced by the accumulation of reactive oxygen species (ROS) [3]. The superoxide anion radical (O_2_**^•^**^−^), the hydrogen peroxide (H_2_O_2_), and the singlet oxygen (^1^O_2_) are the main ROS produced in plant cells, mainly in the light reactions of photosynthesis, but are kept in a homeostasis by the antioxidative enzymatic and non-enzymatic systems [16,17,18,19]. However, under miscellaneous environmental stress conditions, the absorbed light energy surpasses the amount that can be used for photochemistry, developing an increased production of ROS that can cause oxidative stress [20,21,22]. Photoprotective mechanisms preventing ROS formation through the process of non-photochemical quenching (NPQ) or by scavenging ROS are activated to prevent oxidative damage and improve fitness [23,24,25,26,27,28,29,30,31].

The impacts of excess Zn cations or nanoparticles on the photosystem II (PSII) function were recently described mainly in Zn-tolerant or -hyper-tolerant species [3,7,32,33,34,35], and were found to be leaf-age dependent [12]. After both the young and mature leaves of *Arabidopsis thaliana* were exposed to CuZn nanoparticles, PSII function was detected to be negatively influenced in young leaves, while in mature leaves, a beneficial effect on PSII function was observed [36]. *Noccaea caerulescens*, a model species for examining metal tolerance, can accumulate Zn–Cd–Ni at enormously high concentrations in its aboveground tissues. It has been suggested that the reduced plastoquinone (PQ) pool of PSII, through H_2_O_2_ signaling, activated a detoxification mechanism of photosynthetic function to heavy metal stress by inducing heavy-metal vacuolar sequestration [14].

Increased evidence suggests that plants have developed protective strategies to neutralize the side-effects from heavy-metal toxicity and/or mechanisms that allow them to have a better performance under heavy-metal exposure and, thus, can use heavy metals as beneficial elements [37]. A hormetic response of PSII photochemistry was noticed after exposure of *Cymodocea nodosa* to ZnO nanoparticles, with a 24 h exposure time to be required for the induction of an adaptive response mechanism [38]. In clary sage, a hormetic response of PSII function to Cd exposure was observed, with a rapid enhancement of PSII functionality at short time exposure to Cd, and an inhibition at longer duration implying toxicity effects [39]. Recently, the phenomenon of hormesis was shown to occur in a number of organisms independent of the kind of stressor or the physiological process examined [40,41,42,43,44]. Hormesis is frequently described as a biphasic dose–response, with a positive low dose or short time stimulation response, representing an “over-compensation” response to a disruption in homeostasis and a high dose or longer duration inhibition [44,45,46,47,48,49].

Pulse-amplitude-modulated (PAM) fluorescence imaging enables analyses on spatial heterogeneities in photosynthetic parameters and is considered to be a developing tool to evaluate the changes that occur under unfavorable environmental conditions (biotic–abiotic) in the amount of the light energy use at the whole leaf’s surface [50,51,52,53,54,55]. The response of whole-leaf PSII photochemistry of heavy-metal-tolerant plant species has been shown to be heterogenous [14,33,39] and to be related to the whole-leaf spatial accumulation pattern of the heavy metal [56]. Thus, the leaf spatial pattern of high heavy-metal accumulation was correlated with the spatial pattern of the decreased effective quantum yield of PSII photochemistry (Φ*_PSII_*) [56].

Understanding how and where any disturbance in elemental homeostasis of living organisms is taking place has driven the development of laser ablation–inductively coupled plasma–mass spectrometry (LA–ICP–MS) as the principal elemental imaging technique for biologists [57]. LA–ICP–MS imaging is well-suited to quantify elemental accumulation across a wide concentration range and conditions, while also providing data on the spatial distribution in leaf structures to investigate the interaction of natural and anthropogenic pollutants with plants [57]. By coupling the methodologies of chlorophyll fluorescence imaging analysis and LA–ICP–MS, the mechanism of photosynthetic tolerance to heavy-metals stress can be effectively elucidated [56].

The genus *Salvia* L., in the family Lamiaceae, consists of about 1000 species that are used in traditional and conventional medicines [58]. Among them, *Salvia sclarea* L. (clary sage) is cultivated worldwide as an ornamental and essential-oil-bearing plant, being used in the aromatic and food industry [59,60]. Its essential oils present analgesic and antidiabetic, as well as antifungal, antimicrobial, and anti-inflammatory, effects [59,61,62,63]. Recently it has been considered an economically important plant for phytoextraction and phytostabilization of Cd^2+^ and Zn^2+^ contaminated soils [3,7,59,64].

Although hormesis stimulated by heavy metals is a renowned phenomenon, the involved biological mechanisms are not fully understood [65]. Exogenous low-Zn concentrations induced a hormetic dose–response relationship in wheat seedlings and promoted its growth by enhancing the photosynthetic activity [65]. Based on our previous research works on the heavy-metal effects on clary sage [3,7,39,56,64] and other heavy-metal-tolerant plant species [14,32,33], we hypothesized that the PSII response of *Salvia sclarea* to Zn exposure will be a hormetic one if the proper tolerance mechanisms could be activated. In such a case, PSII photochemistry would be regulated by the photoprotective mechanism of NPQ and could resemble a spatiotemporal heterogenous response.

## 2. Results

### 2.1. Zinc and Nutrient Concentrations before and after Zinc Exposure

After 8 days of exposure of *S. sclarea* plants to 900 μM Zn, the concentration of Zn in the aboveground tissues (leaves and shoots) increased by 2-fold (*p* < 0.05), and in the belowground (roots) by 449-fold (Table 1). Zinc treatment did not change Cu in aboveground tissues but increased it in belowground tissues by 12.7-fold, while Mn did not change in belowground tissues but increased by 95% in aboveground tissues (Table 1). Iron increased by 38% in the aboveground tissues and by 4.7-fold in belowground tissues (Table 1). Magnesium was the only nutrient that decreased significantly in both belowground (44%) and aboveground (9%) tissues, while Ca decreased significantly in belowground tissues but increased significantly (22%) in aboveground tissues (Table 1).

### 2.2. Leaf Distribution Pattern of Zn, Ca, and Mg

The accumulation patterns of Zn, Ca, and Mg after 8 days of exposure of *Salvia sclarea* to Zn were assessed by laser ablation–inductively coupled plasma–mass spectrometry (LA–ICP–MS). The distribution of Zn and Ca in the leaf segment of *Salvia sclarea* after 8 days of exposure to Zn presented a similar pattern, with the high concentrations being located in the mid-leaf-vein area (Figure 1a,b). However, after 8 days of exposure to Zn, more Zn was accumulated in the belowground tissues, while in the case of Ca, more was accumulated in the aboveground tissues (Table 1). Magnesium, the only metal that was reduced in both aboveground and belowground tissues after 8 days of exposure to Zn (Table 1), presented a different distribution pattern (Figure 2a) from Zn (Figure 1a) and Ca (Figure 1b), being accumulated mainly at the mesophyll leaf area (Figure 2a). Nevertheless, the accumulation pattern of Mg in the control *Salvia sclarea* plants was similar to Ca in the control plants (Appendix A). Despite the less accumulated Zn in the control plants (Table 1), its accumulation pattern (Appendix A) was similar to 8 days of exposure to Zn (Figure 1a).

At the mid-leaf-vein area with the high Zn content, the lowest fraction of open PSII reaction centers (q*p*) was observed (Figure 2b). In contrast, in the control plants, at the mid-leaf-vein area, the highest fraction of open PSII reaction centers (q*p*) was observed (Appendix A).

### 2.3. Maximum Efficiency of Photosystem II Photochemistry and Light-Energy Utilization in PSII before and after Zinc Exposure

No change in the maximum efficiency of PSII photochemistry (F*v*/F*m*) was observed as a result of the excess Zn exposure up to 3 days at the whole-leaf level (Figure 3a), but after 8 days of exposure of *Salvia sclarea* to Zn, F*v*/F*m* increased significantly at the whole-leaf level (Figure 3a).

The quantum efficiency of the PSII photochemistry (Φ*_PSΙΙ_*) of *S. sclarea* plants exposed to Zn for 2 days remained the same as that of the controls (Figure 3b), but at further Zn exposure (3 and 8 days), it increased (*p* < 0.05), with the highest values observed after 8 days of exposure to Zn (Figure 3b). This increase indicated a higher fraction of absorbed light energy to be directed to photochemistry. After 2 days of exposure to Zn, the highest values in the quantum yield of regulated heat dissipation in PSII (Φ*_NPQ_*) were observed, while further exposure resulted in a gradual decrease of Φ*_NPQ_* (Figure 3c). A significant gradual decrease in the quantum yield of non-regulated energy dissipated in PSII (Φ*_NO_*) was observed in *S. sclarea* exposed to Zn (Figure 3d).

### 2.4. Changes in the Photoprotective Heat Dissipation and the Redox State of the Plastoquinone Pool after Zinc Exposure

The non-photochemical quenching (NPQ) that reflects the photoprotective heat dissipation of excitation energy increased significantly after 2 days of exposure to Zn, and although it decreased at further Zn exposure (3 and 8 days), it still remained higher than that of the control plants (Figure 4a).

The electron transport rate (ETR) followed the pattern of Φ*_PSΙΙ_***,** showing no difference from the control plants after 2 days of exposure to Zn (Figure 4b), but at further Zn exposure (3 and 8 days), it increased significantly, with the highest values observed after 8 days of exposure to Zn (Figure 4b).

The fraction of open PSII reaction centers (q*p*), estimated by the puddle model (Figure 4c) remained the same as that of the controls in plants exposed to Zn for 2 days, but at further Zn exposure, it increased. The fraction of closed PSII reaction centers (1-*qL*)—based on the “lake” model for the photosynthetic unit (Figure 4d)—of *S. sclarea* plants exposed to Zn for 2 days remained the same as that of the controls, but at further Zn exposure, it decreased significantly.

### 2.5. Changes in the Excess Excitation Energy in Photosystem II and the Efficiency of Open PSII Reaction Centers after Zinc Exposure

The excess excitation energy (EXC) increased significantly after 2 days of exposure to Zn, but it decreased significantly at further Zn exposure (3 and 8 days), showing the lower values after 8 days (Figure 5a).

The efficiency of the excitation energy capture by open PSII centers (F*v*′/F*m*′) decreased significantly after 2 days of exposure to Zn, but after 8 days of exposure, it had the highest values (Figure 5b).

### 2.6. The Efficiency of the Oxygen Evolving Complex before and after Zinc Exposure

No change in the efficiency of the OEC on the donor side of PSII (evaluated by the ratio F*v*/F*o*) was observed in response to the excess Zn exposure up to 3 days (Figure 6), but after 8 days of exposure of *S. sclarea* to Zn, the F*v*/F*o* increased significantly (Figure 6).

### 2.7. The Spatial Pattern of the Quantum Yield of PSII Photochemistry (Φ_PSΙΙ_) before and after Zinc Exposure

The effective quantum yield of photochemistry (Φ*_PSII_*) in the control plants showed a spatial heterogeneity pattern, with the higher values observed at the mid-leaf-vein area (Figure 7a). Exposure of clary sage plants to Zn for 8 days resulted in the highest Φ*_PSII_* values at the whole-leaf level (Figure 3b), but in contrast to the control plants (Figure 7a), the lowest values were observed at the mid-leaf-vein area (Figure 7b).

### 2.8. The Hormetic Response Curves of the Quantum Yield of PSII Photochemistry (Φ_PSII_) and the Quantum Yield of Regulated Non-photochemical Energy Loss in PSII (Φ_NPQ_) in Response to Excess Zinc

There was a decrease of the effective quantum yield of PSII photochemistry (Φ*_PSII_*) in *Salvia sclarea* plants after exposure to Zn for 2 days. This effect changed after 3 days of exposure to Zn, with Φ*_PSII_* to increase above the control level and maintained up to 8 days (Figure 8a). This pattern of hormesis corresponds to a U-shaped biphasic response curve (Figure 8a).

Looking at the response of the quantum yield of regulated non-photochemical energy loss in PSII (Φ*_NPQ_*) of *Salvia sclarea* plants after exposure to Zn for 2 days, we observed an increase that maintained up to 3 days above the control level but decreased after 8 days of exposure to Zn (Figure 8b). Thus, an inverted U-shaped biphasic curve describes the Φ*_NPQ_* response of *Salvia sclarea* plants to Zn exposure (Figure 8b).

### 2.9. Leaf Heterogeneity in Phenolic Accumulation

After 8 days of Zn exposure, a heterogenous spatial distribution of phenolics inside leaf cells was noticed (Figure 9b); meanwhile, such a response was never noticed in the control (Figure 9a). The leaf cross-sections revealed that there was an increased osmiophilic material accumulation inside the mesophyll cells surrounding the leaf vein (circle in Figure 9b), and a similar heterogeneity was also noticed in the leaf epidermis (arrow in Figure 9b).

### 2.10. Heterogeneity in Chlorophyll Fluorescence Parameters between Paraveinal Areas and Intercostal Fields

The higher spatial heterogeneity in q*p* (Figure 2b and Appendix A) and Φ*_PSΙΙ_* (Figure 7a,b) was observed between mid-leaf vein-paraveinal tissues and intercostal fields. The mid leaf vein-paraveinal tissues after 8 days Zn treatment possessed significant lower Φ*_PSΙΙ_*, NPQ, and q*p*, values compared to intercostal fields (Appendix A). In contrast, mid leaf vein-paraveinal tissues after 8 days exposure to Zn possessed significant higher Φ_NO_ values compared to intercostal fields (Appendix A). Significant lower NPQ values in mid leaf vein-paraveinal tissues compared to intercostal fields was also detected after 3 days exposure to Zn (Appendix A).

## 3. Discussion

Excessive Zn has been found to (i) induce the production of ROS and stimulate membrane lipid peroxidation; (ii) disturb chlorophyll synthesis and leaf photosynthesis and, thus, inhibit plant growth; and (iii) disturb antioxidant-enzyme activities [65,66,67]. The impact of plant exposure to excess Zn on the photosynthetic process has been frequently studied [3,7,34,35,65], but there is a lack of knowledge on a hormetic effect of Zn on PSII function. Chlorophyll *a* fluorescence measurements have been used successfully to probe the function of the photosynthetic machinery and especially of PSII [14,68,69,70,71].

Elemental imaging sheds light on the fundamental chemical makeup of living organisms [57]. Elucidating the pattern of nutrient distribution in leaves is of great significance in understanding the response mechanism of PSII to heavy-metal stress [56]. In our work the use of LA–ICP–MS enabled leaf Ca, Zn, and Mg mapping, illuminating the response of PSII photochemistry to excess Zn. Considering our results and the recent literature [56,72], we determined ^13^C to be the most appropriate for referencing leaves, since it stabilizes the distributions of Ca, Zn and Mg.

Despite a decreased belowground Ca uptake after 8 days of exposure to Zn, the aboveground Ca content increased significantly under Zn exposure (Table 1), corroborating its role in photoprotection and repair of PSII under environmental stresses [32,73]. On the contrary, the high Zn accumulation in roots (Table 1) is indicative of a transport restriction to the aboveground tissues. Plants can take up Zn from the growth environment via roots and transport it to aboveground tissues [7,65]. However, excessive accumulation of Zn in the aboveground tissues has been reported to disrupt chlorophyll synthesis [7,65,67] and either maintain the efficiency of photosynthesis [67] or inhibit photosynthesis [65]. In contrast to the high Zn accumulation of mid-leaf veins, mesophyll cells accumulated small amounts of Zn (Figure 1a), suggesting an ability to inhibit Zn accumulation in the chloroplast [32,74,75,76], as an adaptation strategy [15,77,78]. Leaf Zn accumulation incorporates an efficient metal import and loading in vacuoles, along with a controlled redistribution from this compartment [15]. Nevertheless, less Zn vacuolar sequestration in the root is an important mechanism for Zn hyperaccumulation [66].

The high co-localization of Zn with Ca (Figure 1a,b) in the mid-leaf veins suggests a possible mechanism of Ca amelioration of the detrimental effects of high Zn content on PSII photochemistry, as judged from the q*p* (Figure 2b) and Φ*_PSΙΙ_* (Figure 7b) values that both decreased in the mid-leaf-vein area, despite their increased values at the whole-leaf level (Figure 2b and Figure 7b). It must also be mentioned that, in the control plants, both Φ*_PSΙΙ_* (Figure 7a) and q*p* (Appendix A) possessed the highest values in the mid-leaf-vein area. The exogenous application of Ca is mentioned to alleviate the adverse effects of abiotic stresses in plants by activating defense responses against environmental stimuli [73,79,80]. Accumulation of phenolic compounds in the cells of the leaf’s mid-vein area (Figure 8b), probably due to the high Zn concentration on this area (Figure 1a), is suggested to perform a main role in coping with the adverse effect of the high Zn concentration that occurred.

Manganese, which is vital in the OEC of PSII for water oxidation (Hill reactions) and produces electrons on the donor side of PSII [81,82], increased significantly in the aboveground tissues of *S. sclarea* plants exposed to Zn for 8 days (Table 1); thus, it could have contributed to the higher efficiency of the OEC (F*v*/F*o*) that was observed to occur at the same time (Figure 6). On the other hand, toxic Mn concentrations can cause a decreased Hill activity [83]. Increased Mn content in leaves has been stated to induce an enhancement of the antioxidant-enzyme activities in the common bean [84], perennial ryegrass [85], and *Populus cathayana* [86].

Iron has a vital role as a component of the enzymes involved in the electron transport process [87,88]. In our experiment, the Fe levels were significantly increased in the aboveground tissues after 8 days of exposure of *S. sclarea* plants to Zn (Table 1), with a concomitant increase in the electron transport rate (ETR) (Figure 4b).

Magnesium is the principal component of the tetrapyrrole ring of chlorophyll molecule [89], and excess Zn application has been documented to decrease chlorophyll content [90]. The decreased chlorophyll content reported for *S. sclarea* plants exposed to 900 μM Zn for 8 days [7] can be attributed to the decreased Mg content in the aboveground tissues that was observed (Table 1). However, if plants had fewer light-harvesting pigments (e.g., chlorophylls and carotenoids), light might be absorbed more judiciously [91]. Downregulating chlorophyll synthesis (e.g., by suppressing the rate of 5-aminolevulinate synthesis) is the best option for optimizing antenna size to increase PSII quantum yield (Φ*_PSII_*) [92]. Thus, a decrease in Mg content results in a decreased chlorophyll content that can protect PSII from photo-oxidative stress [93]. This can explain the reason why the observed Mg-deficiency symptoms occur in plants grown under high light intensity [94].

The main pigments for absorbing the light quanta and transferring the energy to the reaction centers for the charge separation and the consequential electron transport are the chlorophyll molecules [95,96,97]. Mutants that have reduced chlorophyll content retain smaller light-harvesting chlorophyll antenna and a more efficient light partitioning; thus, they can be used to improve photosynthetic efficiency [98]. Improved photosynthetic efficiency is achieved via a better allocation of absorbed light energy that also reduces photo-oxidative stress [99]. Clary sage plants, after exposure to 8 days of Zn, possessed increased F*v*/F*m*, Φ*_PSII_*, and ETR values (Figure 3a,b and Figure 4b). Lower F*v*/F*m* values (Figure 3a) indicate a higher degree of photoinhibition [100,101,102]. Thus, the decreased relative excess energy at PSII (Figure 5a) after the exposure of clary sage plants to 8 days of Zn indicated improvements related to PSII efficiency.

The Zn-treated plants with a lower chlorophyll content compared to the controls are characterized by a reduced probability of ^3^Chl* formation, and this translates to a reduced probability to produce ^1^O_2_ [103], as it was documented by the observed lower Φ*_NO_* (Figure 3d), thus pointing to reduced ROSs [104]. Thus, our results suggest that the light absorbed by Zn-treated plants with reduced chlorophyll content was more efficiently partitioned to photosynthesis [98], and excess Zn can be used to improve photosynthetic efficiency. The decreased chlorophyll content in Zn-treated clary sage plants can be regarded as a mechanism that can lower the photoinhibition and photodamage of PSII [91,92]. The lower efficiency of control sage plants to utilize the absorbed light energy for photochemistry (Φ*_PSII_*) (Figure 3b) or to safely dissipate it as heat (Φ*_NPQ_*) (Figure 3c), resulted in increased Φ*_NO_* (Figure 3d) and, thus, to an increased formation of the triplet chlorophyll state (^3^Chl*) population that produced the reactive ^1^O_2_ [105,106]. An increased Φ*_NO_* reflects the failure of a plant to protect itself, and this failure eventually leads to photodamage [107,108].

The decreased Mg content in the aboveground tissues contributed to decreased chlorophyll content that reduced excess absorption of sunlight and operated to improve Φ*_PSII_* (Figure 3b) and to decrease excess energy at PSII (EXC) (Figure 5a), thus lowering the degree of photoinhibition (F*v*/F*m*) (Figure 3a) and protecting PSII from photo-oxidative stress. According to Genty et al. [109], an increased quantum yield of PSII photochemistry (Φ*_PSII_*) can be ascribed either to an increased fraction of open PSII reaction centers (q*p*), or/and to an increased efficiency of these centers (F*v*′/F*m*′). The molecular mechanism by which the decreased chlorophyll content in Zn-treated leaves resulted in a more efficient use of the absorbed light energy and an improved PSII photochemistry was the increased fraction of open PSII reaction centers (q*p*) (Figure 4c) and, mainly, the increased efficiency of the reaction centers (F*v*′/F*m*′) (Figure 5b) that enhanced ETR (Figure 4b).

It can be postulated that exposure of *Salvia sclarea* plants to Zn resulted in chlorophyll degradation that possibly occurred during the second day of exposure, when an excess light energy at PSII (EXC) (Figure 5a) resulted in an increased oxidative stress, as we also observed in Salvia roots after 2 days of exposure to 900 μM Zn (unpublished data). As a consequence of the oxidative stress, chlorophyll degradation occurred [110].

The cations Fe^2+^, Mg^2+^, Mn^2+^, and Ca^2+^ may have major roles in regulating (directly or indirectly) photosynthetic efficiency [111].

The increased excess energy at PSII (EXC) after 2 days of exposure to 900 μM Zn (Figure 5a) corresponded to an increased oxidative stress observed in Salvia roots (unpublished data). Under unfavorable environmental conditions when the absorbed light energy exceeds the capacity of photochemistry, an increased ROS production occurs that causes oxidative stress [20,21,22]. Under such conditions, the levels of ROSs increase, including O_2_^•−^ and hydrogen peroxide (H_2_O_2_), but singlet oxygen (^1^O_2_) production may not [112]. In accordance with this, the increased EXC at PSII after 2 days of exposure to Zn (Figure 5a) did not coincide with any increase in ^1^O_2_, probed by the quantum yield of non-regulated energy dissipated in PSII (Φ*_NO_*) that decreased gradually with increased exposure time to Zn (Figure 3d). Singlet oxygen (^1^O_2_) can be formed by the triplet chlorophyll state (^3^Chl*) that reacts with O_2_ when PSII is unable to utilize the absorbed light energy for photochemistry (Φ*_PSII_*) or safely dissipate it as heat (Φ*_NPQ_*) [105,106]. The probability of ^1^O_2_ formation can be calculated by Φ*_NO_* [103]. A decreased Φ*_NO_* in Zn-exposed *S. sclarea,* as compared to the controls (Figure 3d), implies a better photoprotection and is indicative of a lower ^1^O_2_ production [103,106]. Phenolic accumulation (Figure 9b) that is considered to contribute to ROS scavenging could also contribute to an ROS decrease [7,55].

Exposure to Zn for 2 days increased the excess energy at PSII (EXC) (Figure 5a), and this was accompanied by a decrease in the effective quantum yield of PSII photochemistry (Φ*_PSII_*) in *Salvia sclarea* plants. This effect changed after 3 days of exposure to Zn, with Φ*_PSII_* increasing above the control level and being maintained up to 8 days (Figure 9). The quantum yield of regulated non-photochemical energy loss in PSII (Φ*_NPQ_*) of *Salvia sclarea* plants after exposure to Zn for 2 days showed an increase that was maintained up to 3 days above the control level, but after 8 days of exposure to Zn, the Φ*_NPQ_* decreased (Figure 8b). The decrease of Φ*_PSII_* in *Salvia sclarea* plants after exposure to Zn for 2 days was overcompensated by the increase of the photoprotective heat dissipation of the regulated non-photochemical energy loss in PSII (Φ*_NPQ_*) (Figure 3c) that resulted in a lowering of Φ*_NO_* (Figure 3d).

Hormesis relies highly on the plant-study strategy, the selection of dose range and the number, and the exposure duration [44,48,52,113,114,115,116]. Thus, PSII hormetic responses can be observed only in appropriately designed studies [44]. A hormetic response of PSII is triggered by the NPQ mechanism that is a strategy to protect the chloroplast from oxidative damage by dissipating excess light energy as heat and preventing the damaging formation of ROS [44]. The increased NPQ at 2 days of exposure to Zn (Figure 4a), induced by the increased production of ROS on the same day, is the possible the mechanism that triggered the hormetic response of PSII [44]. Originally considered as damaging metabolic by-products, ROSs are now documented as an important part of numerous cellular functions and signaling molecules [108,117,118,119].

Future forecasts for population and climate change raise concerns over increased food insecurity [120]. Increasing crop yields to feed the rising human population has generated modern crop cultivars with high yields but low micronutrient contents [121]. Fluctuations of the weather conditions greatly influence plant growth and development, eventually affecting crop yield and quality, as well as plant survival [22,122,123]. Micronutrient deficiencies that are considered as hidden hunger, mostly in Fe and Zn, persist as one of the most severe public health challenges, affecting more than three billion people globally [120]. Deficiencies in Zn and Fe are often linked to the deprived nutritional status of agricultural soils, resulting in low amounts of these nutrients [120]. Zinc deficiency is a severe threat in about one-third of the world’s population, with countries such as India, Pakistan, China, and Turkey facing rigorous Zn-deficiency symptoms [124].

In our study, the exposure of clary sage for 8 days to 900 μM Zn resulted not only in increased Zn content but also in increased Fe content in both aboveground and belowground tissues, with a concomitant hormetic response of PSII functionality. Iron and Zn are the nutrients that are considered essential micronutrients for human health [121]. Thus, the exposure of crops to excess Zn in hydroponics can be regarded as an economical approach to ameliorate these nutritional deficiencies [121].

Foliar application of Zn has been noted to enhance the photoprotective mechanisms in drought-stressed wheat plants [125]. In our experiment, NPQ values were higher after 8 days of exposure to Zn, compared to that of the control (Figure 4a), thus documenting the photoprotective character of the NPQ parameter [126].

Hormesis can usually be exploited as a quantitative measure of biological plasticity through adaptive responses under biotic- or abiotic-stress conditions [48,127]. These adaptive responses can be triggered by exposing plants to low levels of biotic or abiotic stressors that protect plants through the stimulation of cellular defense mechanisms [44,127]. Hormetic/biphasic dose–response relationships were frequently observed in plants accumulating heavy metals (i.e., Cd, As, Cr, Zn, Pb, and Cu), with the topmost stimulation typically being about 30–60% more the control values [128]. The hormetic stimulation is related to each metal, species, tissue, and endpoint studied, but quantitative features of the hormetic dose–response are similar across all (hyper)accumulation studies, with results being highly generalizable and independent of the plant species, endpoints measured, and metals studied [128]. However, underlying specific mechanistic strategies are different between plants and animals [46].

Our results clearly show that *S. sclarea* leaves show a cell/tissue-specific heterogenous response not only in terms of PSII photochemistry (Figure 7) but also in terms of the nutrient-distribution pattern (Figure 1a,b and Figure 2a) and the phenolic leaf tissue accumulation (Figure 8). This heterogeneity in leaf response against biotic and abiotic factors has been established by various researchers [24,33,55,56,129,130] and is again, here, being verified. The findings of this work can be regarded as approaches being deployed to improve Fe and Zn accumulation in the aromatic and medicinal plant clary sage and improve its photosynthesis. Thus, the conclusion drawn from our data with clary sage, which is being used in the aromatic and food industry, can contribute to map the options to achieve food and nutritional security.

## 4. Materials and Methods

### 4.1. Plant Material and Growth Conditions

*Salvia sclarea* L. (clary sage) seeds were surface sterilized in 10% (*v*/*v*) sodium hypochlorite, rinsed in sterile distilled water, and subsequently germinated on soil in a growth room. After one month, the seedlings were transferred for another month in continuously aerated modified Hoagland nutrient solution in pots [3]. The nutrient solution {1.5 mM KNO_3_, 1.5 mM Ca(NO_3_)_2_, 0.5 mM NH_4_NO_3_, 0.25 mM KH_2_PO_4_, 0.5 mM MgSO_4_, 50 μM NaFe(III)EDTA, 23 μM H_3_BO_3_, 5 μM ZnSO_4_, 4.5 μM MnCl_2_, 0.2 μM Na_2_MoO_4_, and 0.2 μM CuSO_4_} renewed completely every 3 days [56,64]. The plants were grown in a growth room under the following conditions: 14/10 h day/night photoperiod, 24 ± 1/20 ± 1 °C day/night temperature, 55 ± 5/65 ± 5% day/night relative humidity, and photosynthetic flux density 200 ± 20 μmol photons m^−2^ s^−1^ [3].

### 4.2. Zinc Treatments

Two-month-old clary sage plants were subjected to treatment with either Hoagland nutrient solution (with 5 μM Zn, as Zn-sufficient, control) or Hoagland nutrient solution plus 900 μM Zn (excess Zn), with Zn supplied as ZnSO_4_ (Appendix A). Nutrient solutions were renewed every 2 days so that nutrient contents would remain constant [3,56]. Each treatment was replicated four times. The photosynthetic efficiency of plants was measured after 2, 3, and 8 days of exposure to Zn.

### 4.3. Zinc and Nutrient Cation Determination by Inductively Coupled Plasma–Mass Spectrometry (ICP–MS)

Five Salvia plants per each treatment, control and 2, 3, and 8 days of exposure to 900 μM Zn, were harvested, separated into belowground (roots) and aboveground (shoots-leaves), washed with deionized water, dried at 65 °C to constant biomass, milled, and finally sieved. Dried samples were digested in a microwave-assisted digestion system, as described previously [39], and analyzed by an inductively coupled plasma–mass spectrometry (ICP–MS) model ELAN DRC II (PerkinElmer Sciex, Toronto, ON, Canada) [131]. ICP–MS operational conditions, instrumental settings, calibration solutions, data validation, and validation parameters were as described before [39]. An analysis was performed for Zn, and for the elements Mn, Ca, Fe, and Cu.

### 4.4. Laser Ablation–Inductively Coupled Plasma–Mass Spectrometry

Salvia leaves were analyzed in vivo, using an ICP–QMS spectrometer (Elan DRC II, Perkin-Elmer Sciex, Guelph, ON, Canada) equipped with a laser ablation system (LA; model LSX-500, CETAC Technologies, Omaha, NE, USA) operating at a wavelength of 266 nm, as described in detail previously [56,131]. The instrumentation was optimized on a daily basis by ablating the standard-reference glass material NIST SRM 610 and adjusting the nebulizer gas flow, RF generator power, and ion lens voltage in order to obtain the maximum signal intensity [56]. The ThO^+^/Th^+^ intensity ratios were always below 0.2%, the doubly charged ions ^42^Ca^2+^/^42^Ca^+^ were < 0.5%, and the ^238^U^+^/^232^Th^+^ intensity ratio was less than 1.2. The instrumental and analytical conditions of LA–ICP–MS are summarized in Appendix A. For bioimage generation, LA-iMageS software was used [132].

### 4.5. Chlorophyll Fluorescence Imaging Analysis

Chlorophyll fluorescence measurements were performed in clary sage control and 2-, 3-, and 8-day Zn-treated plants, using an Imaging-PAM Fluorometer M-Series MINI-Version (Heinz Walz GmbH, Effeltrich, Germany). All leaves were dark-adapted for 20 min before measurement. Chlorophyll fluorescence analysis was performed as described in detail previously [29]. In each leaf, we selected the areas of interest (AOI), and for each AOI, the minimum (F*o*) and the maximum (F*m*) chlorophyll *a* fluorescence in the dark, together with the maximum chlorophyll *a* fluorescence in the light (F*m*′), were measured. The minimum chlorophyll *a* fluorescence in the light (F*o*′) was estimated according to Oxborough and Baker [133] as F*o*′ = F*o*/(F*v*/F*m* + F*o*/F*m*′), where F*v* is the variable chlorophyll *a* fluorescence (in the dark-adapted leaves), calculated as F*m* − F*o*. Steady-state photosynthesis (F*s*) was measured after 5 min of illumination time, with AL of 220 μmol photons m^−2^ s^−1^ corresponding to the growth light of clary sage plants. By using Win software (Heinz Walz GmbH, Effeltrich, Germany), we estimated the chlorophyll fluorescence parameters described in detail in Appendix A.

Representative results of color-coded images are also shown to reveal the whole Salvia leaf PSII spatiotemporal response to excess Zn exposure.

### 4.6. Light Microscopy

Leaf segments from clary sage control and 2-, 3-, and 8-day Zn-treated plants were chemically fixed, sectioned, stained, and observed as reported previously [64].

### 4.7. Statistics

All data were tested for normality with the Shapiro–Wilk test and for homogeneity of variance with Levene’s test prior to statistical analysis [134]. Data were normally distributed, but the assumption of homogeneity was not met. A Welch’s one-way ANOVA was performed to compare the effect of the Zn treatments on each of the chlorophyll fluorescence parameters, followed by post hoc analysis with Games–Howell test. All analyses were performed by using the IBM SPSS Statistics for Windows, version 28.0 Armonk, NY, USA: IBM Corp., released in 2021 [93].

## 5. Conclusions

In this study, we revealed that PSII of clary sage shows a hormetic response and also tolerance to excess Zn exposure, as was evident by the higher F*v*/F*m*, Φ*_PSII_*, ETR, q*p*, and F*v*/F*o*, after 8 days of exposure to Zn (Figure 3a,b, Figure 4b,c and Figure 6). This better PSII function revealed after 8 days of exposure to Zn, was evident by (i) the higher efficiency of the OEC (F*v*/F*o*) that was possible due to the significantly increased Mn; (ii) the increased electron transport rate (ETR) due to the significantly increased Fe in the aboveground tissues; and (iii) the decreased Mg content in the aboveground tissues (Table 1) that possible contributed to the increased PSII quantum yield (Φ*_PSII_*). The higher F*v*/F*m* values after 8 days of exposure to Zn indicated a lower degree of photoinhibition and a higher degree of photoprotection. Thus, we assumed that the decreased Mg content in the aboveground tissues contributed to decreased chlorophyll content that reduced excess absorption of sunlight and operated to improve photosystem II photochemistry (Φ*_PSII_*), decreasing excess energy at PSII (EXC), and lowering the photoinhibition (F*v*/F*m*).

## Figures and Tables

**Figure 1 ijms-23-11232-f001:**
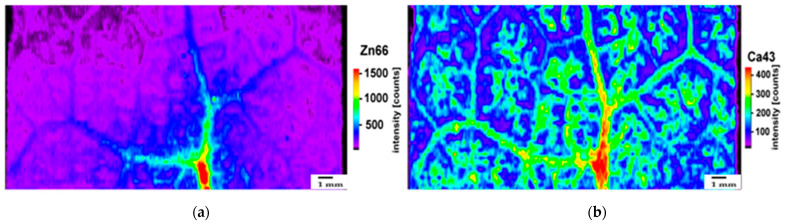
The distribution pattern of Zn (**a**) and Ca (**b**) in a *Salvia sclarea* leaf segment after 8 days of exposure to Zn. Both elements show a high concentration at the same place of the main leaf vein. Zn^66^ and Ca^43^ intensities were normalized by using C^13^. Scale bar: 1 mm.

**Figure 2 ijms-23-11232-f002:**
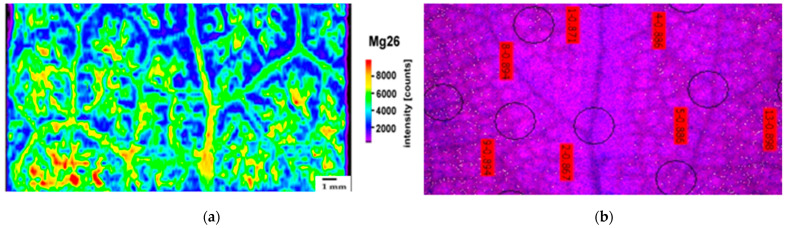
The distribution pattern of Mg in a *Salvia sclarea* leaf segment after 8 days of exposure to Zn (Mg^26^ intensities were normalized by using C^13^) (**a**). The color-coded images of the fraction of open PSII reaction centers (q*p*) of a *Salvia sclarea* leaf segment after 8 days of exposure to Zn, showing the areas of interest (AOI) in circles, together with red labels of the q*p* values, in the main leaf vein and the surrounding area (**b**). Scale bar in (**a**): 1 mm.

**Figure 3 ijms-23-11232-f003:**
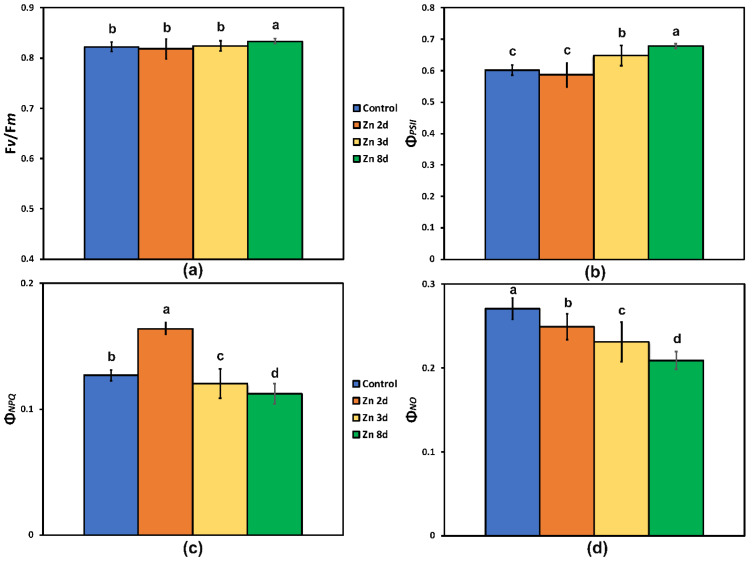
The maximum efficiency of PSII photochemistry (F*v*/F*m*) (**a**); the effective quantum yield of PSII photochemistry (Φ*_PSII_*) (**b**); the quantum yield of regulated non-photochemical energy loss in PSII (Φ*_NPQ_*) (**c**); and the quantum yield of non-regulated energy dissipated in PSII (Φ*_NO_*) (**d**), estimated at 220 μmol photons m^−2^ s^−1^ actinic light (AL) intensity, after exposure of clary sage plants to 0 (control), 2, 3, and 8 days of 900 μM Zn. Columns (error bars ± SD, *n* = 5) with different lowercase letters are statistically different (*p* < 0.05).

**Figure 4 ijms-23-11232-f004:**
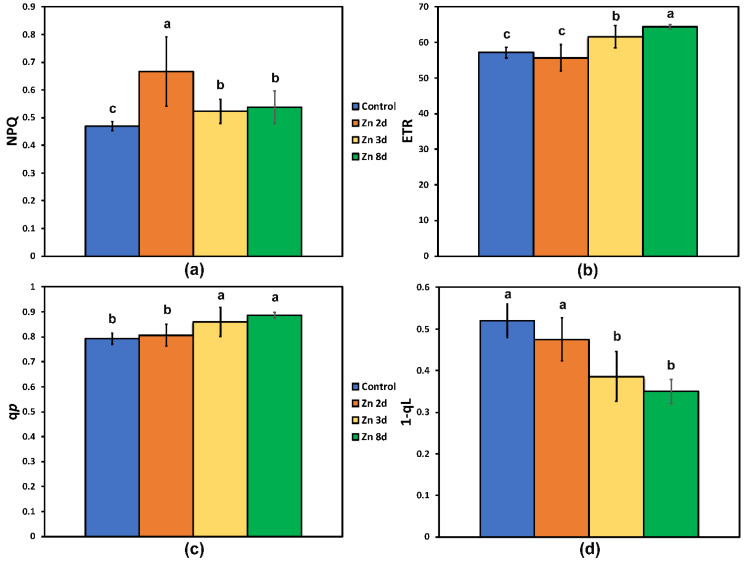
The non-photochemical quenching that reflects heat dissipation of excitation energy (NPQ) (**a**); the electron transport rate (ETR) (**b**); the fraction of open PSII reaction centers (q*p*), a measure of the redox state of quinone A (Q_A_) (**c**); and the fraction of PSII reaction centers in closed state (based on a “lake” model for the photosynthetic unit) (**d**), estimated at 220 μmol photons m^−2^ s^−1^ AL intensity after exposure of clary sage plants to 0 (control), 2, 3, and 8 days of 900 μM Zn. Columns (error bars ± SD, *n* = 5) with different lowercase letters are statistically different (*p* < 0.05).

**Figure 5 ijms-23-11232-f005:**
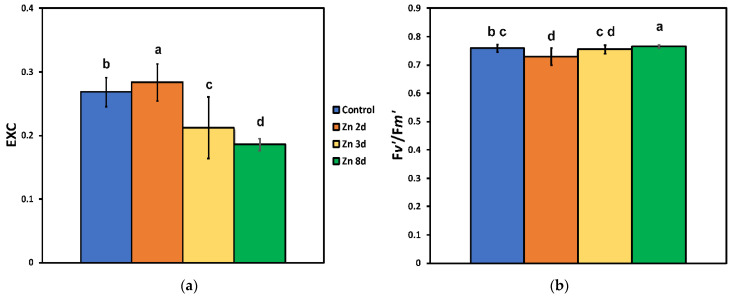
The relative excess energy at PSII (EXC) (**a**) and the efficiency of excitation energy capture by open photosystem II reaction centers (F*v*′/F*m*′) (**b**), estimated at 220 μmol photons m^−2^ s^−1^ actinic light (AL) intensity, after exposure of clary sage plants to 0 (control), 2, 3, and 8 days of 900 μM Zn. Columns (error bars ± SD, *n* = 5) with different lowercase letters are statistically different (*p* < 0.05).

**Figure 6 ijms-23-11232-f006:**
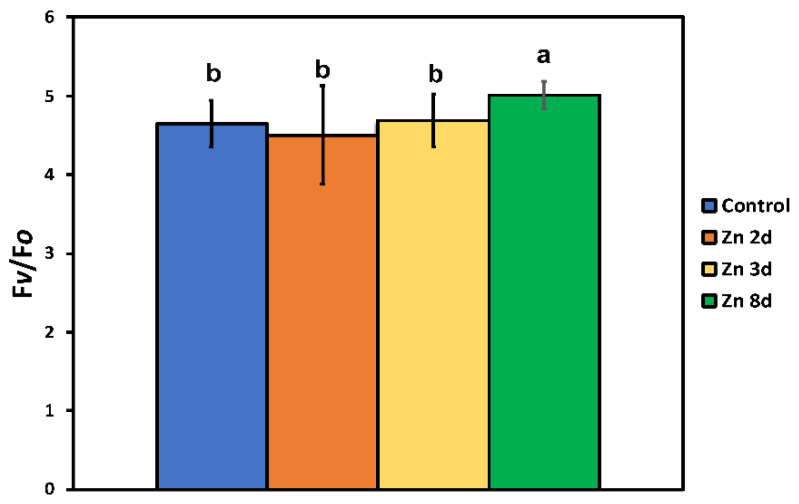
The efficiency of the oxygen evolving complex (OEC) on the donor side of PSII (F*v*/F*o*) after exposure of clary sage plants to 0 (control), 2, 3, and 8 days of 900 μM Zn. Columns (error bars ± SD, *n* = 5) with different lowercase letters are statistically different (*p* < 0.05).

**Figure 7 ijms-23-11232-f007:**
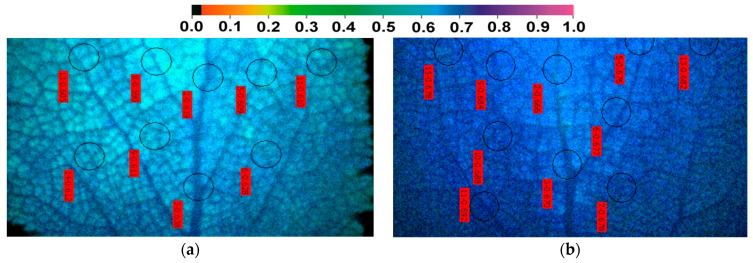
Representative color-coded leaf-segment pictures of the effective quantum yield of PSII photochemistry (Φ*_PSΙΙ_*), under 220 μmol photons m^−2^ s^−1^ actinic light (AL), corresponding to the growth light of the control (**a**) and after exposure to 900 μM Zn for 8 days (**b**). The areas of interest (AOIs) are shown in circles with the corresponding Φ*_PSΙΙ_* values. The color code shown at the top ranges from values 0.0 to 1.0.

**Figure 8 ijms-23-11232-f008:**
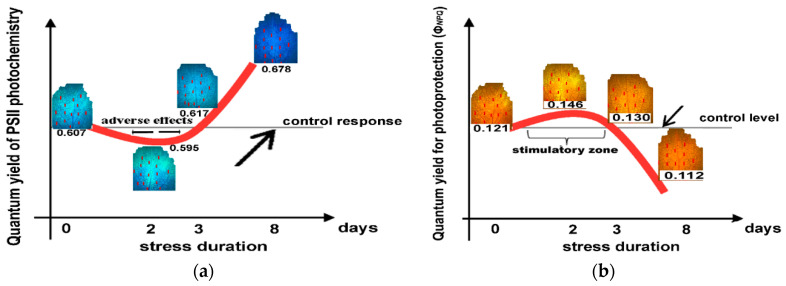
A U-shaped biphasic response curve of the effective quantum yield of PSII photochemistry (Φ*_PSII_*) after exposure of *Salvia sclarea* plants to Zn, with a short time (up to 2 days) of exposure to decrease Φ*_PSII_*, a longer exposure time (3 days) to enhance Φ*_PSII_* due to the induction of stress defense response, and a further exposure time (8 d) to enhance further Φ*_PSII_* (**a**). An inverted U-shaped biphasic response curve of the quantum yield of regulated non-photochemical energy loss in PSII (Φ*_NPQ_*) after exposure of *Salvia sclarea* plants to Zn, with a stimulatory zone duration (up to 3 days) exposure, and an inhibition of Φ*_NPQ_* at longer exposure time (**b**).

**Figure 9 ijms-23-11232-f009:**
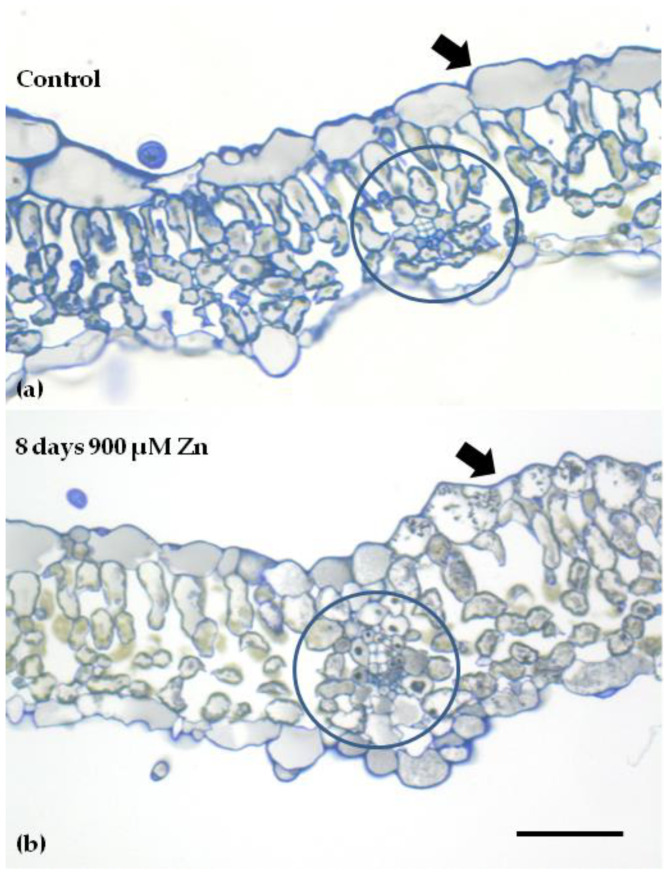
Leaf cross-sections of control (**a**) and after 8 days of 900 μM Zn exposure (**b**). Heterogeneity in osmiophilic material accumulation is noticed after exposure to Zn. Dark material is being accumulated both in mesophyll cells surrounding the vein (circle) and on the right side of the epidermis (arrow) in the Zn-treated leaves, (**b**) but not in the control (arrow) (**a**). Scale bar: 50 μm.

**Table 1 ijms-23-11232-t001:** The content of Zn, Mn, Ca, Fe, Cu, and Mg, expressed as µg g^−1^ dry weight (DW), in control and 8-day Zn-exposed *Salvia sclarea* plants. Mean values (±SD).

Treatment	Plant tissues	Zn	Mn	Ca	Fe	Cu	Mg
Control	aboveground	88.1 ± 2.6	37.2 ± 1.1	8531 ± 252	77.2 ± 2.3	11.6 ± 0.3	4377 ± 131 *
	belowground	89.3 ± 2.7	40.8 ± 1.2	6214 ± 186 *	561 ± 17	20.8 ± 0.6	3273 ± 107 *
Zn 8d	aboveground	175.9 ± 5.3 *	72.6 ± 2.2 *	10,394 ± 301 *	106.5 ± 3.2 *	13.1 ± 0.4	3982 ± 119
	belowground	40,060 ± 1202 *	37.5 ± 1.1	5874 ± 176	2643 ± 78 *	264.5 ± 7.9 *	1835 ± 55

* An asterisk indicates statistically significant difference (*p* < 0.05) between control and 8 days of Zn treatment for the same plant tissues.

## Data Availability

The data presented in this study are available in this article.

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
