# Peer review of "A Hormetic Spatiotemporal Photosystem II Response Mechanism of Salvia to Excess Zinc Exposure"

_ijms, 2022, doi:10.3390/ijms231911232_

Round 1

Reviewer 1 Report

-a Photo of plants Zn treated an control should be supplied.

- the table 1. needs revision please check the value pf SD for copper content “0,62” probably you meant “0.62” however I would avoid commas used (it’s only suggestion) – I think that simply 40 060 would be better than 40,060. Additionally, assurance should be the same within the same metal data.

- Fig 1. and Fig 8– scale bars are needed

- Fig 2b the readable of this figure should be improved

Author Response

Comments and Suggestions for Authors

-a Photo of plants Zn treated an control should be supplied.

A photo of control and Zn treated plants was supplied as supplementary Figure.

- the table 1. needs revision please check the value pf SD for copper content “0,62” probably you meant “0.62” however I would avoid commas used (it’s only suggestion) – I think that simply 40 060 would be better than 40,060. Additionally, assurance should be the same within the same metal data.

Yes, we corrected the copper content to “0.6”. We deleted commas and adopted your suggestions.

- Fig 1. and Fig 8– scale bars are needed

Scale bar was included in the legend of Figure 1 and in Fig. 8, that is now Fig. 9.

- Fig 2b the readable of this figure should be improved

We tried to improve Fig. 2b.

Author Response

The article titled: A hοrmetic spatiotemporal Photosystem II response mechanism of Salvia to excess zinc exposure by Moustakas et al. is centered on studying the the specific biological mechanism involved in zinc-induced hormesis in plants.

The researchers have shown that excess zinc exposure for 8 days increases calcium, iron, manganese and zinc concentrations in the aboveground tissues but decreases magnesium concentration in the above ground as well as below ground tissues. With increased zinc exposure, there was tissue-specific accumulation of zinc and calcium in the mid vein areas of leaves while magnesium accumulated mainly in the mesophyll leaves. The researchers have studied various PSII chlorophyll fluorescence parameters using chlorophyll fluorescence imaging (PAM) and elemental imaging (LA-ICP-MS). The experimental shows that with a high exposure to zinc for 8 days lead to an increase in FV /FM , FV’ /F M’, FV /F o , φPSII , and ETR and qP. The fraction of closed PSII reaction centers decreased significantly with prolonged zinc exposures. φ NPQ and the excess excitation energy increased after 2 days of exposure of zinc but decreased upon further exposures. φ NO gradually decreased with prolonged exposures to zinc

Although this research has contributed to the understanding of how zinc-induced hormesis can affect PSII function via regulated non-photochemical quenching, the article has some weaknesses. The authors are advised to correct these flaws to improve the quality of the manuscript and scientific soundness.

Thank you for your critical comments that helped us to improve our manuscript. The manuscript was revised considering your comments.

The major flaws are:

  1. The article does not show the distribution pattern of zinc and calcium in control leaves in Figures 1 & 2. Figure 8 is also missing the control. The controls must be shown along with experimental samples.

The distribution patterns of zinc, calcium, and magnesium in control leaves was included as supplementary Figures. A control leaf cross section for Figure 8 (now Figure 9) was added along with the 900 μM Zn one.

  1. The increases/decreases in the different chlorophyll fluorescence parameters are small in magnitude in many cases, and cannot be accurately estimated from the bar graphs in Figures as sometimes error bars overlap for some samples. Hence it would be nice to show all numerical values along with SD in supplementary Tables so readers have the exact numbers and can comprehend clearly the exact magnitude of changes in the different chlorophyll fluorescence parameters.

The error bars overlap for many samples and for this reason extensive statistical analysis was performed and the results are shown as lowercase letters so readers can have the exact magnitude of changes in the different chlorophyll fluorescence parameters.

  1. Figure 2b red labels and the numerical values in these labels cannot be read clearly by readers. White fonts should be used for labeling. Figure S1 label values could be read but to keep uniformity, if the authors use white fonts in Figure 2b, they should also use white fonts in Figures S1.

The colour figures with the red labels and the black values are produced by the PAM software automatically so they cannot be changed without interfering in the picture. Otherwise, we could be blamed for changing the original figures. Thus, we tried to improve Figure 2b and Figure 7.

  1. The authors have made 2 assumptions in the article: 1. The function of calcium in alleviating toxic effects of zinc and 2) ROS is inducing NPQ after 2 days of exposure to excess zinc. No experimental evidences have been presented. Hence the speculation about the protective function of Calcium and the link between ROS, NPQ and hormesis can be discussed only in the discussion section and not in the abstract. In an abstract typically main findings from performed experiments are presented.

We have eliminated these two assumptions from the Abstract and we shortened the discussion in the Discussion section.

5.There is no Figure 6b. Then why label Figure 6 as Figure 6a? This figure should be labeled as Figure 6 and not as Figure 6a.

Thank you for pointing this. It was corrected.

  1. In the abstract in line #26, it is stated the decreased magnesium content led to lowering of photoinhibition (FV/FM). This phrasing is incorrect as an increase in FV/FM means lowering of photoinhibition.

We rephrased this sentence (line 26) to be clear: “lowering the degree of photoinhibition, as judged from the increased Fv/Fm ratio.”

  1. Fig 1 should also have a color intensity code like shown in Figure 2.

Yes, it should have. A color intensity code was included in the first manuscript but due to Figure editing by the journal team it was lost. We provide it now again and hope that any Figure editing will not change the Figure. We also provide the revised manuscript in pdf format.

  1. In Figure 7: line # 247 should read as: From left to right (and not right to left) exposure of-

Yes, it was wrong, thank you for pointing it. However, we changed the Figure by eliminating 2- and 3-days exposure to Zn, so as the values to be readable in the remaining pictures of the control and 8-days exposure to Zn, that was the main goal of this Figure.

  1. Discussion section line #s 341-360: Smaller chlorophyll antenna simply means ratio of Chla/Chlb is higher and it does not denote overall decrease in chlorophyll content. Truncated chlorophyll antenna leads to higher solar energy absorption efficiency and superior photosynthetic productivity. Please read the cited research articles in your manuscript carefully. The authors have over-simplifies chlorophyll reduction and have linked it to superior photosynthetic performance. The authors are strongly advised to correctly present the scientific information in these sentences.

We have rewritten this section using almost exactly the sentences of the cited articles.

Other minor issues:

  1. Authors should break up long sentences into short sentences. Because of the usage of long sentences which are combining multiple parameters together, the scientific results (as shown in the figures) are not being described correctly or the English becomes very confusing. For example: see lines 210 - 214; lines 240 - 243. There are numerous such cases throughout the text. please rephrase these sentences so it is a smooth reading for readers.

We break up long sentences or rephrased others in order to be “smooth reading for readers”.

  1. Redundant information throughout the text. The introduction and discussion section need to be condensed. For example: 69 - 73 is are redundant as the same info is presented in lines 60-69. Numerous such redundancies exist throughout the text.

We eliminated the redundant information in lines 69–73 and throughout the text (e.g., lines 281-284 in the old manuscript).

  1. line # 93: stressor should be replaced with stress

We believe that the word stressor is a better choice in this sentence and it was also used in the original citations. 

4.Table S2: φ NO should read as quantum yield of nonregulated energy loss in PSII

Yes, we corrected it. Thank you for pointing it.

  1. qp should be stated as qP.

We have used the symbol qp in many published articles and we would like to keep it also in this one.

  1. Authors are highly recommended to have their manuscript edited by an English proficient person/editor.

We corrected the English language.

Round 2

Reviewer 2 Report

The authors have taken into consideration all the suggestions and have addressed all concern. This has improved the quality of the manuscript.

Author Response

Thank you for your critical comments that helped us to improve our manuscript.
